# Fast Model Selection and Hyperparameter Tuning for Generative Models

**DOI:** 10.3390/e26020150

**Published:** 2024-02-09

**Authors:** Luming Chen, Sujit K. Ghosh

**Affiliations:** Department of Statistics, North Carolina State University, Raleigh, NC 27695, USA; sujit.ghosh@ncsu.edu

**Keywords:** integral probability metric, hypothesis testing, Maximum Mean Discrepancy, multi-armed bandits, generative adversarial networks

## Abstract

Generative models have gained significant attention in recent years. They are increasingly used to estimate the underlying structure of high-dimensional data and artificially generate various kinds of data similar to those from the real world. The performance of generative models depends critically on a good set of hyperparameters. Yet, finding the right hyperparameter configuration can be an extremely time-consuming task. In this paper, we focus on speeding up the hyperparameter search through adaptive resource allocation, early stopping underperforming candidates quickly and allocating more computational resources to promising ones by comparing their intermediate performance. The hyperparameter search is formulated as a non-stochastic best-arm identification problem where resources like iterations or training time constrained by some predetermined budget are allocated to different hyperparameter configurations. A procedure which uses hypothesis testing coupled with Successive Halving is proposed to make the resource allocation and early stopping decisions and compares the intermediate performance of generative models by their exponentially weighted Maximum Means Discrepancy (MMD). The experimental results show that the proposed method selects hyperparameter configurations that lead to a significant improvement in the model performance compared to Successive Halving for a wide range of budgets across several real-world applications.

## 1. Introduction

The performance of the generative models depends heavily on so-called hyperparameters which include the model architecture, the choice of training objective, regularization and training algorithms. However, the choice of these hyperparameters is often problem-dependent, and it is unknown a priori which configuration would produce the best results in terms of a specific distance or divergence measure. With the rich set of objective functions and training algorithms proposed in recent years and the growing number of tuning parameters associated with them, it is crucial to develop computationally efficient search methods for hyperparameter configurations that yield models with a desired performance within a fixed budget constraint.

The problem of efficient model search and hyperparameter optimization has recently been dominated by Bayesian optimization approaches, e.g., [1,2,3], which speed up the search for good configurations by modeling the underlying structure of the search space. These approaches select and evaluate hyperparameter configurations in an adaptive manner, trying to find good configurations faster than baselines such as random search or grid search. While Bayesian optimization is efficient in tuning few hyperparameters, its efficiency often degrades significantly when the search dimension becomes much higher. Wang et al. [4] showed that for high-dimensional problems, standard Bayesian optimization methods perform similarly to random search. Moreover, traditional Bayesian optimization methods (that are often based on Gaussian processes) can only work on continuous hyperparameters, but not categorical ones (e.g., the choice of the training objective). The vast majority of these hyperparameter selection procedures consider the training of machine learning models to be black-box procedures, and only evaluate models after they have been trained to convergence. It thus seems natural to ask the following question: *Can we terminate some of these poor-performing hyperparameter settings early to speed up hyperparameter optimization?* This is one of the primary questions that we address in this work.

In fact, there is a line of research that perceives hyperparameter optimization as an adaptive computational resource allocation problem, where the type of resources can be iterations, execution time, data samples, or even total money to spend with a cloud computing provider. These approaches evaluate partially trained models and make decisions on the fly, allocating more resources to promising hyperparameter configurations while early stopping those that are not. They allow for the training of multiple models simultaneously. And by quickly eliminating unpromising ones and paralleling the model training for different hyperparameter configurations, more hyperparameter configurations can be examined. Swersky et al. [5], Domhan et al. [6] and Klein et al. [7] made parametric assumptions on the convergence behavior of learning curves to devise early stopping rules. However, these assumptions are strong and restrictive for the kinds of learning curves that are typically found in training machine learning models. In contrast, Sparks et al. [8] cast it as a multi-armed bandit problem, viewing each hyperparameter configuration as an ‘arm’, and resources constrained by some predetermined budget are allocated among them by some heuristic rule. Jamieson and Talwalkar [9] and Li et al. [10] studied a similar problem but in the non-stochastic setting and based their resource allocation strategies on the Successive Halving algorithm originally proposed in Karnin et al. [11] for stochastic settings.

However, most existing methodologies on fast model selection and hyperparameter tuning, including all of those discussed above, mainly focus on supervised learning tasks where the data are labeled and the performance of the partially trained models is represented by their losses on a hold-out test set. However, for unsupervised learning tasks, such as many generative modeling tasks that are performed to generate high-quality samples, e.g., [12], fast model search approaches, though important, have been largely scarce in the literature. In light of the rapidly growing literature on generative modeling, it is important to be able to perform efficient model search and hyperparameter tuning based on the quality of samples generated from the candidate models.

In this work, we focus on the best hyperparameter configuration identification problem for generative models. We perceive it as an adaptive resource allocation problem with a given budget (e.g., iteration and training time). We build our approach upon the non-stochastic multi-armed bandit formulation proposed by Jamieson and Talwalkar [9], which makes minimal assumptions on the convergence behavior of the model performance during the training process. Note that, in this work, we restrict the term “Generative Models” to represent models that learn the underlying probability distribution in input data to generate similar samples. To evaluate the performance of generative models, we compute a sample-based distance metric between the samples generated from partially trained models and those from the reference distribution. Particularly, we base our evaluation criterion on the Maximum Mean Discrepancy (MMD) [13] that measures the closeness of the generated samples to the reference distribution. By incorporating statistical tests between partially trained models into the evaluation process, our method effectively identifies the poor-performing configurations early on and allocates more resources to promising configurations.

The remainder of this paper is organized as follows. In Section 2.1, we review the non-stochastic best-arm identification problem and the Successive Halving algorithm. In Section 2.2 and Section 2.3, we present the proposed *Adaptive Successive Halving algorithm* (**AdaptSH**) and provide intuition for it through an example. In Section 3, we present empirical results comparing AdaptSH with Successive Halving. We conclude with a discussion in Section 4.

## 2. Methods

In this section, we present the AdaptSH algorithm. We start with a brief review of the non-stochastic best-arm identification problem and Successive Halving. We subsequently introduce our choice of the metric for comparing the intermediate performance of generative models to decide which models should be trained further, and we provide intuition for our choice via a simple example. We then introduce our proposed statistical test and how we incorporate it into Successive Halving to help us distinguish between candidate models and make early stopping decisions.

### 2.1. Non-Stochastic Best-Arm Identification and Successive Halving

The non-stochastic best-arm identification problem, originally proposed in Jamieson and Talwalkar [9], considers a very general setting that encompasses the hyperparameter optimization problem of interest. It only assumes that the sequence of the losses of each arm (hyperparameter configuration) eventually converges without making any assumptions on the rate of the convergence, monotonicity, or smoothness of the sequence. Hence, it is generally applicable to a wide variety of problems including minimizing a non-convex objective using stochastic gradient descent or some other iterative algorithms. Let *K* denote the total number of arms and let $\ell_{k,j}$ denote the validation error of the *k*th arm after training for *j* units of resources (e.g., iterations). For all k∈{1,2,…,K}, assume $\nu_{k}=\lim_{j\to \infty }\ell_{k,j}$ exists. The goal is to identify $\arg\min_{k}\nu_{k}$ when the resources are constrained by some predetermined budget. Successive Halving, shown in Algorithm 1, is proposed by Jamieson and Talwalkar [9] to solve the above problem. The strategy of Successive Halving follows its name: given a set of *K* arms and a budget B, it splits the given budget evenly across $\log_{2}\left(K\right)$ elimination rounds, uniformly allocates the resources to remaining arms at each round, evaluates their intermediate performance, throws out the worst half until one arm remains. By the design of the algorithm, it allocates exponentially more resources to more promising configurations.
**Algorithm 1** Successive Halving**Input**: Budget B, *K* models $M_{1},\ldots ,M_{K}$1:$S_{0}=\left\{1,2,\ldots ,K\right\}$2:Initialize i=03:n=K4:**while** B>0 **and** n>=2 **do**5:    Allocate $r_{i}=\left\lfloor \frac{B}{n\left\lceil \log_{2}\left(n\right)\right\rceil }\right\rfloor$ units of resource to each model in $S_{i}$6:    $R_{i}=\sum_{j=0}^{i}r_{j}$7:    Sort the intermediate losses of the models in $S_{i}$ such that $\ell_{\sigma_{i}\left(1\right),R_{i}}\leq \ell_{\sigma_{i}\left(2\right),R_{i}}\leq \ldots \leq \ell_{\sigma_{i}\left(n\right),R_{i}}$, where $\sigma_{i}\left(\cdot \right)$ is a bijection from {1, 2, …, n} to $S_{i}$8:    $S_{i+1}=\left\{\sigma_{i}\left(j\right)\;|\;1\leq j\leq \left\lfloor \frac{n}{2}\right\rfloor \right\}$9:    $B=B-nr_{i}$10:  $n=\lfloor \frac{n}{2}\rfloor$11:  i=i+112:**end while**

### 2.2. Exponentially Weighed Average of ${MMD}^{2}$


While in Successive Halving, half of the configurations are discarded at each elimination round, it is not entirely clear why we should do so. Indeed, it is not clear what proportion to discard in each round would lead to better results without prior knowledge about the convergence behavior of the sequences of losses. We propose to use statistical tests to detect when two models have separated in their performance to make on-the-fly elimination decisions. Before we jump into the statistical test, we first introduce our choice of metric to represent the intermediate performance of generative models, based on which we develop our statistical test to distinguish between their model performance.

While for supervised learning tasks, models are usually compared by their validation errors on a hold-out set, there is no such straightforward measure for generative model comparisons. Although there are a number of evaluation measures for generative models that have been proposed in recent years including the average log-likelihood, different variants of the Wasserstein distance [14], Fréchet Inception Distance (FID) [15] and Maximum Mean Discrepancy (MMD), there is no consensus as to which measure best captures the strengths and limitations of generative models and should be used for a fair model comparison. Indeed, there are a number of desired properties for a good measure, including the ability to distinguish generated samples from real ones, favoring models that generate diverse samples, and having low computational and sample complexity. It is unlikely that a single measure can cover all aspects. Since different applications require different trade-offs among the desired properties, it has been argued that the evaluation metric should be chosen with respect to specific applications [16]. On the other hand, previous works have shown through empirical studies that MMD performs well in terms of the discriminability, robustness and efficiency compared to other metrics when it operates in the feature space [17,18]. Moreover, the empirical estimate of MMD enjoys favorable statistical properties such as asymptotic normality, making it a favorable choice to construct two-sample and three-sample tests that compares probability distributions. Therefore, we base our model selection criterion on MMD.

MMD is a metric of probability measures which falls within the family of integral probability metrics (IPMs) [19]. For two probability measures, P and Q, over $X\subset R^{d}$, IPM is defined as
(1)$$D_{F}\left(P,Q\right)=\underset{f\in F}{\sup}E_{P}f\left(X\right)-E_{Q}f\left(Y\right),$$
which is the maximum difference between the mean function values on the two probability measures. The choice of the witness function class F determines the probability metric. The MMD is defined as the IPM with F being the unit ball in a reproducing kernel Hilbert space (RKHS) H, with a positive definite kernel k(·,·):X×X→R,
$$\mathrm{MMD}\left(P,Q;H\right)=\underset{f\in {H,∥f∥}_{H}\leq 1}{\sup}E_{P}f\left(X\right)-E_{Q}f\left(Y\right).$$
It can be interpreted as the distance between the mean embeddings of P and Q into H. It can be shown that the square of the MMD can be expressed as
(2)$${\mathrm{MMD}}^{2}\left(P,Q;H\right)=E_{P⊗P}\left[k\left(X,X^{\prime }\right)\right]-2E_{P⊗Q}\left[k\left(X,Y\right)\right]+E_{Q⊗Q}\left[k\left(Y,Y^{\prime }\right)\right],$$
where *X* and $X^{\prime }$ are independent random variables having distribution P, and *Y* and $Y^{\prime }$ are independent random variables having distribution Q [13]. It immediately follows that MMD has a straightforward unbiased empirical estimator:(3)$$\begin{array}{cc} {\mathrm{MMD}}_{u}^{2}\left(X_{m},Y_{n},H\right) & =\frac{1}{m(m-1)}\sum_{i=1}^{m}\sum_{j\neq i}^{m}k\left(x_{i},x_{j}\right)+\frac{1}{n(n-1)}\sum_{i=1}^{n}\sum_{j\neq i}^{n}k\left(y_{i},y_{j}\right)\\ & -\frac{2}{mn}\sum_{i=1}^{m}\sum_{j=1}^{n}k\left(x_{i},y_{j}\right), \end{array}$$
where $X_{m}:=\left\{x_{1},\ldots ,x_{m}\right\}$ and $Y_{n}:=\left\{y_{1},\ldots ,y_{n}\right\}$ are i.i.d. samples from P and Q, respectively. Let $v_{i}:=\left(x_{i},y_{i}\right)$, i=1,…,m be i.i.d samples from P×Q, when m=n. Then,
(4)$${\mathrm{MMD}}_{u}^{2}\left(X_{m},Y_{m},H\right)=\frac{1}{\left(m)(m-1\right)}\sum_{i\neq j}^{m}h\left(v_{i},v_{j}\right)$$
is a U-statistic with $h\left(v_{i},v_{j}\right)=k\left(x_{i},x_{j}\right)+k\left(y_{i},y_{j}\right)-k\left(x_{i},y_{j}\right)-k\left(x_{j},y_{i}\right)$. According to the properties of U-statistics, $\sqrt{m}\left({\mathrm{MMD}}_{u}^{2}\left(X_{m},Y_{m},H\right)-{\mathrm{MMD}}^{2}\left(P,Q,H\right)\right)$ converges weakly to a Gaussian distribution as m→∞, when P≠Q and $E\left(h^{2}\right)<\infty$. MMD and the Wasserstein distance are two extremes of the Sinkhorn divergences, an entropic regularized variant of the Wasserstein distance, e.g., [20,21].

Instead of simply using the ${\mathrm{MMD}}^{2}$ at the current iteration to represent the intermediate performance of each candidate model, we consider an exponentially weighted average of ${\mathrm{MMD}}^{2}$ that takes into account the model performance at previous training iterations, i.e.,
(5)$${\tilde{\ell }}_{k,R}^{\beta ,h}=\frac{\sum_{r=0}^{h-1}\beta^{r}{\mathrm{MMD}}^{2}\left(P,Q_{k}^{R-r},H\right)}{\sum_{r=0}^{h-1}\beta^{r}},\;\;\mathrm{for}\;\;k=1,\ldots ,K\;\mathrm{and}\;R>h-1,$$
where h≥1 is the window size, β∈(0,1) is the decay rate, P is the reference/target distribution and $Q_{k}^{R}$ denotes the distributions of the samples generated from model *k* after being trained for *R* units of resources. An exponentially weighted average smooths the learning curves. As a result, the models’ performance represented by the smoothed ${\mathrm{MMD}}^{2}$ are more distinguishable from each other. Figure 1 shows an illustrative example. Two Generative Adversarial Network (GAN) models with different training objectives are trained on the Half Moons dataset, respectively (details of the dataset can be found in Section 3). At the early stage of training, the variation of the loss tends to be large. While any sequences of losses (under the convergence assumption) would eventually stabilize and be separated from each other even without smoothing, smoothed loss is able to distinguish between the two models at a much earlier stage, which can help us make the right decision about which models should be trained further.

### 2.3. Adaptive Successive Halving with Hypothesis Testing

Now, we introduce our proposed algorithm called Adaptive Successive Halving (AdaptSH), shown in Algorithm 2. As the name of our algorithm suggests, instead of following a predetermined elimination schedule as in Successive Halving, we base our decision on test results and adaptively change the elimination schedule based on the remaining budget and remaining number of arms. In particular, the major difference between AdaptSH and the original Successive Halving lies in line 8 of Algorithm 2, where we perform a sequence of statistical tests to compare the intermediate performance of the current “best” arm and each of other remaining arms by taking samples from them and compare their relative similarity to the target distribution. The corresponding *p*-values are adjusted using the procedure proposed by Benjamini and Yekutieli [22], which controls the false discovery rate. To avoid inflating the type I error rate, we use two independent samples from each model to sort the models (line 7) and to perform the statistical tests (line 8), respectively. The algorithm stops allocating further resources to models that perform significantly worse than the current “best” model, as measured by a desired loss criteria (e.g., MMD). It should be noted that our proposed algorithm can perhaps be used if an alternative loss (e.g., the Wasserstein distance and alike) is chosen, but developing an asymptotic theory required by the statistical tests can be challenging.
**Algorithm 2** Adaptive Successive Halving**Input**: Budget B, *K* models $M_{1},\ldots ,M_{K}$, decay rate β, window size *h*, significance level α1:$S_{0}=\left\{1,2,\ldots ,K\right\}$2:Initialize i=03:n=K4:**while** B>0 **and** n>=2 **do**5:    Allocate $r_{i}=\left\lfloor \frac{B}{n\left\lceil \log_{2}\left(n\right)\right\rceil }\right\rfloor$ units of resource to train each model in $S_{i}$6:    $R_{i}=\sum_{j=0}^{i}r_{j}$7:    Sort the intermediate losses of the models in $S_{i}$ such that ${\tilde{\ell }}_{\sigma_{i}\left(1\right),R_{i}}^{\beta ,h}\leq {\tilde{\ell }}_{\sigma_{i}\left(2\right),R_{i}}^{\beta ,h}\leq \ldots \leq {\tilde{\ell }}_{\sigma_{i}\left(n\right),R_{i}}^{\beta ,h}$, where $\sigma_{i}\left(\cdot \right)$ is a bijection from {1, 2, ..., n} to $S_{i}$8:    Compare $M_{\sigma_{i}\left(1\right)}$ against $M_{\sigma_{i}\left(j\right)}\left(j=2,\ldots ,n\right)$ using three-sample tests comparing the relative closeness of their generated samples to the validation dataset with Benjamini and Yekutieli [22] correction to obtain the adjusted *p*-values $p_{2}^{\mathrm{adjusted}},\ldots ,p_{n}^{\mathrm{adjusted}}$9:    $S_{i+1}=\left\{\sigma_{i}\left(j\right)\;|\;2\leq j\leq n\;\mathrm{and}\;p_{j}^{\mathrm{adjusted}}>\alpha \right\}\cup \left\{\sigma_{i}\left(1\right)\right\}$10:  $B=B-nr_{i}$11:  $n=|S_{i+1}|$12:  i=i+113:**end while**

The statistical test in line 8 is a three-sample relative similarity test that aims to determine with high significance whether the samples generated by the current “best” model are closer to the evaluation data set than those of each remaining model. While there is rich literature on two-sample test problems for multivariate data, statistical tests for three-sample relative similarity are rarely studied in the literature. Bounliphone et al. [23] propose a relative similarity test:$$\begin{array}{c} H_{0}:{\mathrm{MMD}}^{2}\left(P,Q,H\right)={\mathrm{MMD}}^{2}\left(P,T,H\right)\\ H_{1}:{\mathrm{MMD}}^{2}\left(P,Q,H\right)<{\mathrm{MMD}}^{2}\left(P,T,H\right), \end{array}$$
which tests the null hypothesis that two distributions Q and T are equally close to a target distribution P against the alternative hypothesis that Q is closer to P than T. They propose the following test statistic:(6)$${\mathrm{MMD}}_{u}^{2}\left(X_{m},Y_{m},H\right)-{\mathrm{MMD}}_{u}^{2}\left(X_{m},Z_{m},H\right),$$
where $X_{m}:=\left\{x_{1},\ldots ,x_{m}\right\}$, $Y_{m}:=\left\{y_{1},\ldots ,y_{m}\right\}$ and $Z_{m}:=\left\{z_{1},\ldots ,z_{m}\right\}$ are iid samples from P, Q and T, respectively. The test statistic is asymptotically Gaussian, which directly follows from Hoeffding [24] (Theorem 7.1), which states that the joint distribution of several U-statistics converges weakly to a multivariate Gaussian distribution as m→∞.

We generalize their test to compare the averages of ${\mathrm{MMD}}^{2}$ between two arms. In Section 2.2, we defined the exponentially weighted ${\mathrm{MMD}}^{2}$, ${\tilde{\ell }}_{k,R}^{\beta ,h}$ that we use to represent the performance of arm *k* after being trained for *R* units of resources. In particular, we want to determine with high significance whether one arm has a smaller exponentially weighted ${\mathrm{MMD}}^{2}$ than the other, which translates to the following null and alternative hypothesis:$$\begin{array}{c} H_{0}:\sum_{r=0}^{h-1}\beta^{r}{\mathrm{MMD}}^{2}\left(P,Q_{1}^{R-r},H\right)=\sum_{r=0}^{h-1}\beta^{r}{\mathrm{MMD}}^{2}\left(P,Q_{2}^{R-r},H\right)\\ H_{1}:\sum_{r=0}^{h-1}\beta^{r}{\mathrm{MMD}}^{2}\left(P,Q_{1}^{R-r},H\right)<\sum_{r=0}^{h-1}\beta^{r}{\mathrm{MMD}}^{2}\left(P,Q_{2}^{R-r},H\right), \end{array}$$
where ${\left\{Q_{i}^{R}\right\}}_{R>0},i=1,2$ denote two sequences of distributions corresponding to two arms that are being compared. It is then natural to use the following test statistic:(7)$$T_{m}^{h,\beta }=\sum_{r=0}^{h-1}\beta^{r}{\mathrm{MMD}}_{u}^{2}\left(X_{m},Y_{m}^{1,R-r},H\right)-\sum_{r=0}^{h-1}\beta^{r}{\mathrm{MMD}}_{u}^{2}\left(X_{m},Y_{m}^{2,R-r},H\right),$$
where $X_{m}:=\left\{x_{1},\ldots ,x_{m}\right\}$, $Y_{m}^{1,R}:=\left\{y_{1}^{1,R},\ldots ,y_{m}^{1,R}\right\}$ and $Y_{m}^{2,R}:=\left\{y_{1}^{2,R},\ldots ,y_{m}^{2,R}\right\}$ are iid samples from P, $Q_{1}^{R}$ and $Q_{2}^{R}$ respectively. The following theorem states the asymptotic normality of the joint distribution of multiple unbiased estimators of ${\mathrm{MMD}}^{2}$s, which follows directly from Hoeffding [24] (Theorem 7.1).

**Theorem 1.** 
*Assume that $E_{v,v^{\prime }∼P\times Q_{i}^{r}}h^{2}\left(v,v^{\prime }\right)<\infty$ and $P\neq Q_{i}^{r}$ for i=1,2 and r=R−h+1,…,R, then*

(8)
$$\sqrt{m}\left(\left(\begin{array}{c} {\mathrm{MMD}}_{u}^{2}\left(X_{m},Y_{m}^{1,R},H\right)\\ \ldots \\ {\mathrm{MMD}}_{u}^{2}\left(X_{m},Y_{m}^{1,R-h+1},H\right)\\ {\mathrm{MMD}}_{u}^{2}\left(X_{m},Y_{m}^{2,R},H\right)\\ \ldots \\ {\mathrm{MMD}}_{u}^{2}\left(X_{m},Y_{m}^{2,R-h+1},H\right) \end{array}\right)-\left(\begin{array}{c} {\mathrm{MMD}}^{2}\left(P,Q_{1}^{R},H\right)\\ \ldots \\ {\mathrm{MMD}}^{2}\left(P,Q_{1}^{R-h+1},H\right)\\ {\mathrm{MMD}}^{2}\left(P,Q_{2}^{R},H\right)\\ \ldots \\ {\mathrm{MMD}}^{2}\left(P,Q_{2}^{R-h+1},H\right) \end{array}\right)\right)\overset{d}{⟶}N\left(0_{2h},\Sigma_{m}^{h,\beta }\right),$$

*where $0_{2h}$ denotes a vector of zeros with length 2h, and $\Sigma_{m}^{h,\beta }$ denotes the covariance matrix.*


The explicit form of $\Sigma_{m}^{h,\beta }$ and its empirical estimate are given in Appendix A. Then, the *p*-value can be approximated by
(9)$$p\approx \Phi \left(\frac{T_{m}^{h,\beta }}{\sqrt{\frac{1}{m}\beta_{h}^{T}\Sigma_{m}^{h,\beta }\beta_{h}}}\right),$$
where $\beta_{h}={\left(1,\beta^{2},\ldots ,\beta^{h-1},-1,-\beta^{2},\ldots ,-\beta^{h-1}\right)}^{T}$ and Φ(·) is the cumulative distribution function of a standard normal distribution. When h=1, our proposed test reduces to the three-sample relatively similarity test proposed in Bounliphone et al. [23]. Notice that as we have a closed form expression to compute the *p*-values using the asymptotic distribution (as given in the Appendix A), there is not much additional overhead in using the statistical tests within our proposed AdaptSH compared to the traditional SH.

We use the same example as we used in Section 2.2 to illustrate the effect of using exponentially weighted average on the results of the statistical test between arms. We apply the test based only on the current ${\mathrm{MMD}}^{2}$ and our proposed test respectively to the two models shown in Figure 1. The tests are performed every 10 iterations, and the alternative hypothesis considered is that Model 1 (represented in blue in Figure 1) has smaller (exponentially weighted) ${\mathrm{MMD}}^{2}$s. The resulting *p*-values are shown in Figure 2. Suppose the significance level α=0.01 is considered. Then, a *p*-value less than 0.01 indicates that the Model 2 is significantly worse than Model 1 and will be stopped from further training based on our model search algorithm. And a *p*-value greater than 0.99 indicates that Model 2 is significantly better then Model 1 and that Model 1 will be early stopped, since it is equivalent to a *p*-value less than 0.01 if the opposite alternative hypothesis were considered. Figure 3 shows that increasing *h* increases the ease of making the right decision during the training process. For this particular example, when h≥7, the chance of early stopping in favor of the worse model reduces to zero.

## 3. Experimental Results

In this section, we compare our proposed algorithm to Successive Halving on two hyperparameter optimization problems for GAN models. In particular, we consider a number of GAN models that are trained using different variants of the Sliced Wasserstein distance. The space of the models to search includes the set of distance metrics to search over and the reasonable ranges of their associated hyperparameters. We consider seven different variants of the Sliced Wasserstein distance and a set of different combinations of hyperparameters for each of them, which sum up to 30 GAN models in total. The details of the hyperparameter configurations considered for each distance metric can be found in Appendix B. The same generator and discriminator architectures are used for all 30 models (See Appendix B).

To evaluate the different search algorithms’ performance under different budgets, we set the total budget of the iterations to a sequence of values, and for each budget let the search algorithms decide how to allocate it amongst the different arms. The final performance of the models is represented by their final losses, defined as $\nu_{k}=\lim_{j\to \infty }\ell_{k,j}$ for each model *k*. As $\nu_{k}$ is unknown, and we approximate it by the loss after training the model for some finite units of resources *R*. In real-world applications, *R* is often determined by the maximum amount of resources that one wants to allocate to any given configuration, which are often inferred from previous training experiences on similar tasks or determined by the time and money one wants to spend on training one model for a particular task. To address the oscillation around the optimal value when the models are trained by stochastic optimization methods, we use the average loss within a small window [R,R+Δ) to approximate $\nu_{k},k=1,\ldots ,K$, i.e, $\hat{v_{k}}=\frac{1}{\Delta }\sum_{j=R}^{R+\Delta -1}{\hat{\ell }}_{k,j}$.

***2-D Half Moons*** We first apply the searching algorithms to compare the 30 GAN models being trained on the Half Moons dataset. We uniformly sample 1000 points on the half moons as the training set and another 1000 points as the validation set. These sample sizes are used for just for numerical illustration but the results are not sensitive to these choices unless we use very small sample sizes. The training data are shown in Figure 4. At each elimination round, 500 points are sampled from each model to compute ${\mathrm{MMD}}_{u}^{2}$ that is used for sorting, and another 500 points are sampled for statistical testing. We set one unit of resources to 10 training iterations. And we vary the budget from 400 to 1400 units by a step of 50. And for each budget, we record the estimated final loss of the selected arm by each searching algorithm. We repeat the experiment for 20 trials. For each searching algorithm and each budget, we compute the average loss of the 20 trials, ${\hat{\mu }}_{i,B}=\frac{1}{20}\sum_{j=1}^{20}{\hat{v}}_{k(i,B,j)}^{j}$, where *i*, *B* and *j* denote the search algorithm, budget and trial number, respectively, k(i,B,j) denotes the index of the model selected by algorithm *i* using budget *B* at trial *j*, and ${\hat{v}}_{k}^{j}$ denotes the estimated final loss of the model *k* at trial *j*. The tuning parameters for the exponential weighting are chosen as β=0.9 and h=6. For the choice of the kernel function in MMD, we follow Bounliphone et al. [23] and use a Gaussian kernel with bandwidth selected as the median pairwise distance between data points. But one can use alternative kernel functions (e.g., the semantic-aware deep kernel proposed by Gao et al. [25]). Additional numerical results for other tuning parameter settings are provided in Appendix C.

As is shown in Figure 5, for the majority of the budgets considered, our proposed method selects arms with smaller final losses on average. To further compare the losses of the arms selected by the two searching algorithms, we conduct a Mann–Whitney U test between the two samples ${\left\{{\hat{v}}_{k(1,B,j)}^{j}\right\}}_{j=1}^{20}$ and ${\left\{{\hat{v}}_{k(2,B,j)}^{j}\right\}}_{j=1}^{20}$ for each B with the alternative hypothesis that the final loss obtained by AdaptSH is stochastically smaller than that of Successive Halving. The *p*-values of the tests are shown in Figure 5. For budgets over 600, the *p*-value stays around 0.05, which indicates that our algorithm outperforms SH with a high confidence for a large range of budgets.

***3-D Swiss Roll*** We next compare AdaptSH to Successive Halving on the Swiss Roll dataset. A total of 1000 points are sampled on the Swiss Roll as the training set and another 2000 points as the validation set. The training data are shown in Figure 6. At each elimination round, 1000 points are sampled from each model to compute ${\mathrm{MMD}}_{u}^{2}$ that is used for sorting, and another 1000 points are sampled for statistical testing. The experiment setting is the same as that in the previous example, as well as the choice of the tuning parameters for the exponential weighting. We repeat the experiment for 20 trials. And the results are shown in Figure 7. Our proposed method selects arms with smaller final losses on most of the budgets, and the advantage is significant for most moderate budgets.

## 4. Conclusions

We have presented a method for hyperparameter optimization for generative models. We cast it as a non-stochastic best-arm identification problem with a fixed budget and identify the clearly underperforming arms early on through statistical tests between average empirical ${\mathrm{MMD}}^{2}$s of partially trained models. In our experiments, we showed that our procedure leads to a significant improvement in the performance of the selected configurations compared to Successive Halving on a wide range of budgets and is robust to the choice of the tuning parameters. Our method does not make any assumptions about the parametric form of the learning curves, nor about their monotonicity or smoothness, which makes it general enough to be suitable for most hyperparameter optimization tasks.

## Figures and Tables

**Figure 1 entropy-26-00150-f001:**
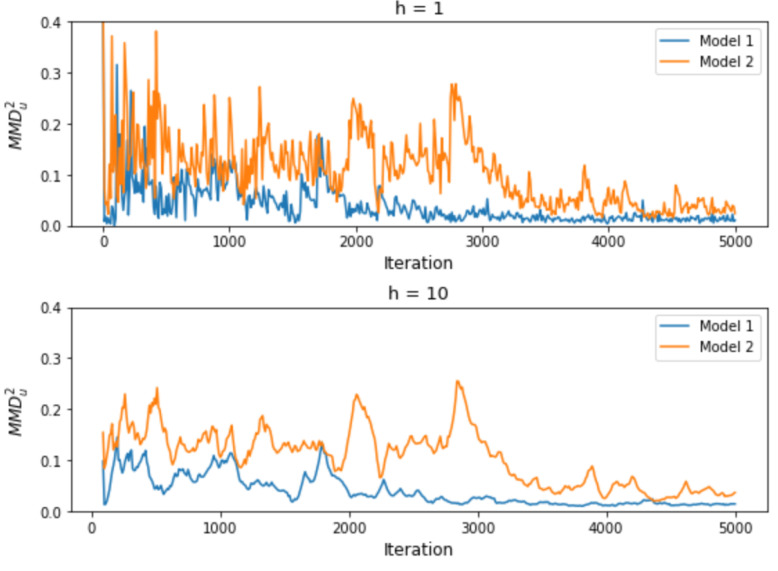
**Upper**: ${\mathrm{MMD}}_{u}^{2}$ versus training iteration; **lower**: exponentially weighted average of ${\mathrm{MMD}}_{u}^{2}$ with h=10 and β=0.9 versus training iteration.

**Figure 2 entropy-26-00150-f002:**
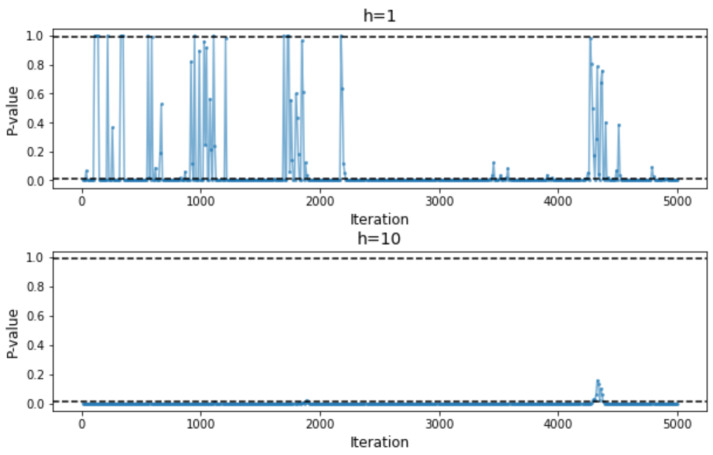
**Upper**: *p*-values of the statistical tests when only the most recent ${\mathrm{MMD}}^{2}$ is considered; **lower**: *p*-values of the statistical tests when historical ${\mathrm{MMD}}^{2}$s within the moving window with h=10 and β=0.9 are considered. The two horizontal dashed lines correspond to *p*-values of 0.01 and 0.99 respectively.

**Figure 3 entropy-26-00150-f003:**
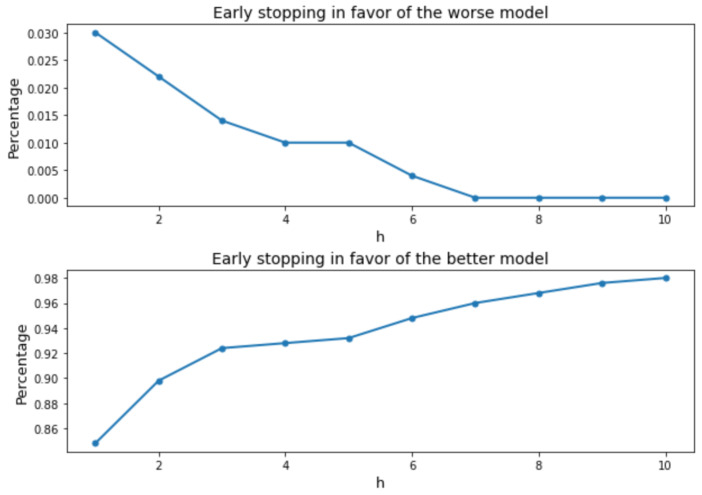
**Upper**: The change of the percentage of tests that have a *p*-value less than 0.01 with *h*; **lower**: The change of the percentage of tests that have a *p*-value greater than 0.99 with *h*.

**Figure 4 entropy-26-00150-f004:**
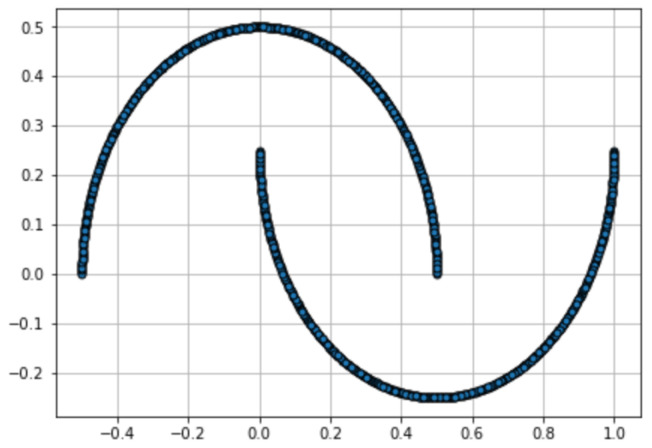
One thousand points sampled on the “Half Moons”.

**Figure 5 entropy-26-00150-f005:**
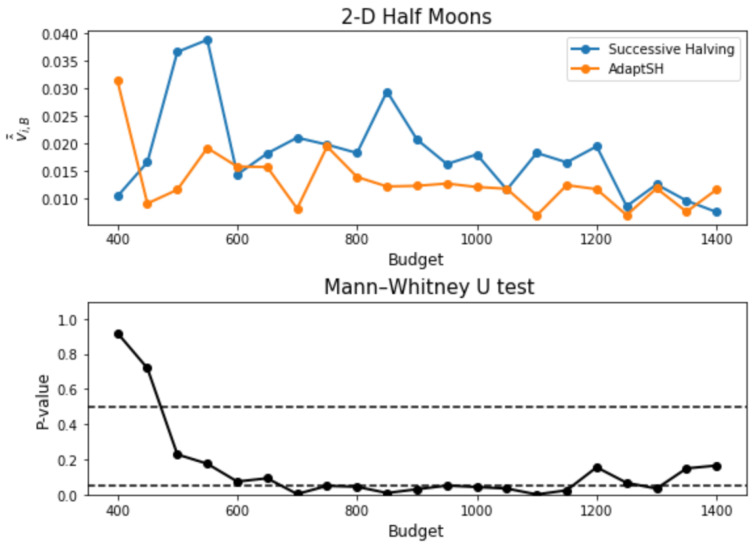
Results on the 2-D Half Moons dataset. **Upper**: Final loss of the selected models averaged over 20 trials against the input budget. **Lower**: *p*-values of the Mann–Whitney U tests. Two horizontal dashed lines represent *p*-value = 0.05 and 0.5, respectively.

**Figure 6 entropy-26-00150-f006:**
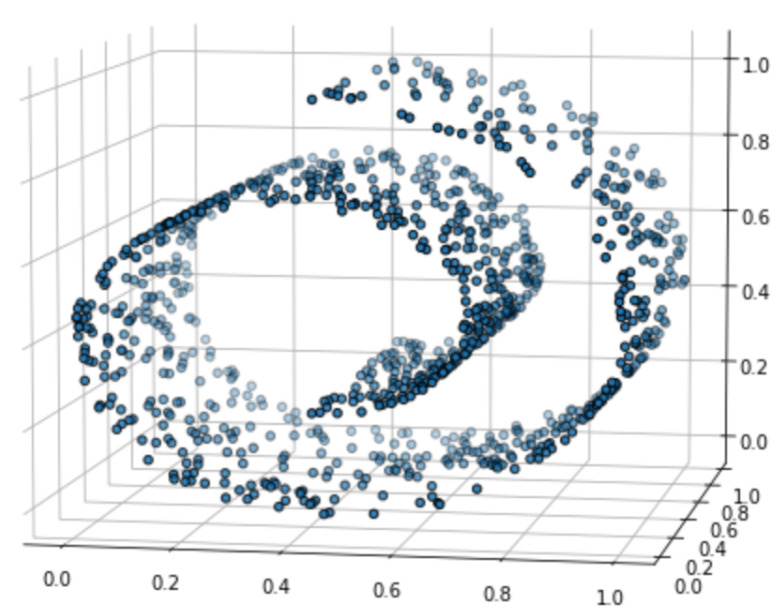
One thousand points sampled on the “Swiss Roll”.

**Figure 7 entropy-26-00150-f007:**
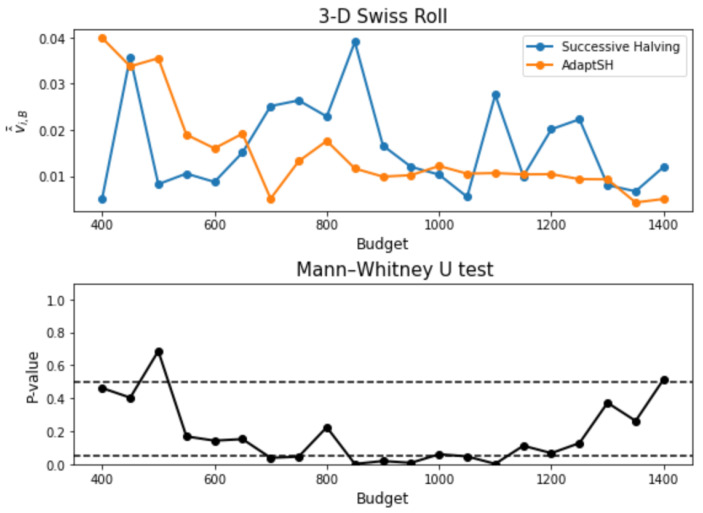
Results on the 3-D Swiss Roll dataset. **Upper**: Final loss of the selected model averaged over 20 trials against the input budget. **Lower**: *p*-values of the Mann–Whitney U tests. Two horizontal dashed lines represent *p*-value = 0.05 and 0.5 respectively.

## Data Availability

Datasets used in this paper were generated during the study. The code to generate the datasets is available upon request.

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
