# Peer review of "Fast Model Selection and Hyperparameter Tuning for Generative Models"

_entropy, 2024, doi:10.3390/e26020150_

Round 1

Reviewer 1 Report

Comments and Suggestions for Authors

General summary

This paper describes how the method of successive halving of Jamieson et al. (2016) can be adapted to unsupervised learning that exploits generative models. The authors rightly point out that the method of successive halving includes a parameter of ½ that controls the selection of the models to be explored in the next round. The authors propose to use a statistical test for this aim. This would be relatively straightforward if the setting was supervised learning.

However, as the aim is to design a method applicable to unsupervised learning problems, the authors proposed a solution that involves samples of the given distribution, and the distributions generated by different models, which are considered as “arms” in so-called “multiarmed-bandit problems”. The problem of selecting models (“arms”) is then cast as a problem of comparing samples and estimating the distances between them. The authors propose to use the Maximum Mean Discrepancy (MMD) distance, and Section 2 provides various technical details. The loss calculated on the validation data is then used to sort the models in each round.

The authors show that it is useful to apply exponentially weighted average of MMDs which provides a smoother measure, which in turn is used to calculate the p-value. The p-value is used to determine whether a given model in the ordered list should (or should not) be passed to the next round.

The authors have evaluated the proposed method on two different problems, 2-D Half Moons dataset and 3-D Swiss-Roll dataset. They show that the loss achieved by the proposed adaptive method of successive halving (AdaptSH) achieves a lower loss than the original SH method for any budgets.

The proposed method represents a significant contribution in the field which can serve as the basis for further work in this domain and further improvements. 

Questions and comments to consider when improving the paper

The description of using the three-sample test could be improved. Step 8 does not mention it. If I understood if correctly, the aim is to compare three samples: The training sample, the sample in rank 1 and sample in position n. The aim is to determine whether the sample in position n is as similar to the training sample, as is sample in position 1. Is this so?

Step 5 deals with allocation of resource to each model in S_i. What exactly does this represent here? Is the size of the sample increased?

What are the overheads of using statistical tests on samples of 1000s points? If it were significant, it should be considered as an “overhead” and used to decrease the available budget B. Otherwise, it could be argued that the proposed method has an unfair advantage wrt.  the successive halving method.

According to your description in Section 3. the number of models considered was 30. This means that SH would need only 5 iterations to identify the best model. Or did I miss something? What was the number of iterations of your proposed method? What does the best model look like? Is there any intuitive justification for the parameter settings identified?

The initial sample size of 1000 was used in both of your problems. What would be the effect if this was set to a different value?

Reviewer 2 Report

Comments and Suggestions for Authors

This paper addresses the selection of an optimal set of hyperparameters, a key issue in the performance of generative models, which is a process that can be notably time-consuming. To expedite this hyperparameter search, the focus of this paper is on adaptive resource allocation. This approach involves swiftly terminating underperforming candidates while allocating increased computational resources to more promising ones based on their intermediate performance assessments.

The method for hyperparameter search is conceptualized as a non-stochastic best-arm identification problem. In this framework, resources such as iterations or training time, limited by a predetermined budget, are distributed among different hyperparameter configurations. The paper proposes a novel procedure that combines hypothesis testing with Successive Halving. This procedure is designed to facilitate resource allocation and early-stopping decisions. It evaluates the intermediate performance of generative models based on their exponentially weighted Maximum Means Discrepancy (MMD).

Experimental findings indicate that the proposed method is adept at selecting hyperparameter configurations that substantially enhance model performance. This improvement is observed across various real-world applications and budget ranges, showing a marked advancement over traditional methods like Successive Halving. The results underscore the efficacy of the proposed approach in optimizing generative model performance through efficient hyperparameter tuning.

In general, this paper presents a novel and significant contribution to the field of generative models, with practical implications that could benefit a wide range of applications. The quality of writing supports the effective communication of these contributions, making the paper a valuable addition to existing literature.

Author Response

Thank you very much for taking the time to review this manuscript. We are very glad to hear your positive remarks, and as there are no additional questions, we have not prepared any separate responses. The major changes in the revised manuscript are indicated with the blue color. We hope that you find the revised version satisfactory and acceptable for publication.